# Guinea Pigs Are Not a Suitable Model to Study Neurological Impacts of Ancestral SARS-CoV-2 Intranasal Infection

**DOI:** 10.3390/v17050706

**Published:** 2025-05-15

**Authors:** Jonathan D. Joyce, Greyson A. Moore, Christopher K. Thompson, Andrea S. Bertke

**Affiliations:** 1Translational Biology, Medicine and Health, Virginia Polytechnic Institute & State University, Blacksburg, VA 24061, USA; jjoyce84@vt.edu; 2Center for Emerging Zoonotic and Arthropod-Borne Pathogens, Virginia Polytechnic Institute & State University, Blacksburg, VA 24061, USA; 3Biomedical and Veterinary Science, Virginia Maryland College of Veterinary Medicine, Virginia Polytechnic Institute & State University, Blacksburg, VA 24061, USA; 4School of Neuroscience, Virginia Polytechnic Institute & State University, Blacksburg, VA 24061, USA; 5Population Health Sciences, Virginia Maryland College of Veterinary Medicine, Virginia Polytechnic Institute & State University, Blacksburg, VA 24061, USA

**Keywords:** SARS-CoV-2, USA-WA1/2020, COVID-19, neuroinvasion, trigeminal ganglia, superior cervical ganglia, dorsal root ganglia, peripheral nervous system, autonomic nervous system, guinea pig

## Abstract

Neurological symptoms involving the central nervous system (CNS) and peripheral nervous system (PNS) are common complications of acute COVID-19 as well as post-COVID conditions. Most research into these neurological sequalae focuses on the CNS, disregarding the PNS. Guinea pigs were previously shown to be useful models of disease during the SARS-CoV-1 epidemic. However, their suitability for studying SARS-CoV-2 has not been experimentally demonstrated. To assess the suitability of guinea pigs as models for SARS-CoV-2 infection and the impact of SARS-CoV-2 infection on the PNS, and to determine routes of CNS invasion through the PNS, we intranasally infected wild-type Dunkin-Hartley guinea pigs with ancestral SARS-CoV-2 USA-WA1/2020. We assessed PNS sensory neurons (trigeminal ganglia, dorsal root ganglia), autonomic neurons (superior cervical ganglia), brain regions (olfactory bulb, brainstem, cerebellum, cortex, hippocampus), lungs, and blood for viral RNA (RT-qPCR), protein (immunostaining), and infectious virus (plaque assay) at three- and six-days post infection. We show that guinea pigs, which have previously been used as a model of SARS-CoV-1 pulmonary disease, are not susceptible to intranasal infection with ancestral SARS-CoV-2, and are not useful models in assessing neurological impacts of infection with SARS-CoV-2 isolates from the early pandemic.

## 1. Introduction

For nearly 200 years, guinea pigs (*Cavia porcellus*) have served as small animal models of neurological, ocular, gastrointestinal, genitourinary, and pulmonary infectious diseases [1]. Given the anatomical and physiological similarities between human and guinea pig lungs, guinea pigs have been used as transmission models for important respiratory pathogens, including *Mycobacterium tuberculosis* (tuberculosis), *Legionella pneumophila* (Legionnaires disease), and influenza [1,2,3]. Guinea pigs have also been used as small animal models for the study of neurotropic viruses including Zika virus, Junin virus, and multiple herpesviruses (herpes simplex virus 1, herpes simplex virus 2; varicella zoster virus) [4,5,6,7]. During the SARS-CoV-1 epidemic of 2002–2004, early research into the virus showed that guinea pigs were susceptible to infection and developed clinical signs and pulmonary pathologies similar to those observed in humans [8,9].

During the SARS-CoV-2 pandemic, early studies assessing the basic virology and pathobiology of the virus were primarily conducted using various transgenic mouse lines created for studying SARS-CoV-1 after the epidemic [10]. These transgenic mice were readily infected with SARS-CoV-2 and developed both pulmonary and neuroinvasive infections. Neuroinvasive disease in these mice was aggressive, leading to rapid decline and death within one week, a complication not observed in humans so soon after infection [11]. The high expression of the transgene encoding the human angiotensin-converting enzyme 2 (hACE2), the host cell protein used by SARS-CoV-1 and SARS-CoV-2 for attachment and entry, was a driver of this severe neuroinvasive disease. The expression of hACE2 at levels exceeding those found in humans and in tissues that are devoid of expression in humans artificially exacerbated virus spread and disease progression within the mice.

As the SARS-CoV-2 pandemic evolved, wild-type golden Syrian hamsters became widely used as a SARS-CoV-2 animal model, as variants gained the ability to bind alternative mammalian ACE proteins, decreasing dependence on transgenic hACE2 models [12,13,14,15]. The K18-hACE2 transgenic mouse model, developed by Stanley Perlman following the SARS-CoV-1 epidemic, expresses hACE2 under the control of the human cytokeratin 18 (K18) promoter [10]. This promoter drives expression primarily in epithelial tissues, including the respiratory tract, a primary target for SARS-CoV-2 infection. These mice are highly susceptible to SARS-CoV-2, developing robust viral replication in the lungs and, in severe cases, viral dissemination to other organs such as the brain. The K18-hACE2 model has been widely used to study SARS-CoV-2 pathogenesis, host responses, and therapeutic interventions; however, the severe disease exhibited during infection makes them an inferior model to golden Syrian hamsters [11,16,17,18,19]. Also, during the pandemic, the impact of SARS-CoV-2 on the nervous system became increasingly noted, with more than 80% of individuals with acute COVID-19 reporting neurological symptoms and over 30% with post-COVID conditions reporting continuation of symptoms. These symptoms impact both the central nervous system (CNS; e.g., headache, anosmia, confusion) and the peripheral nervous system (PNS; e.g., altered sensation/pain, weakness, autonomic dysfunction) [20,21,22,23].

Since guinea pigs have traditionally been used to study both respiratory and neurotropic viruses, including SARS-CoV-1, and since SARS-CoV-2 is a respiratory virus associated with neurological symptoms and sequelae, we aimed to evaluate their potential as a model for assessing SARS-CoV-2 neuroinvasion of the PNS using the ancestral SARS-CoV-2 isolate USA-WA1/2020. Our current study focuses on USA-WA1/2020, rather than contemporary Omicron variants, as the estimated 79.2 million COVID-19 cases from 2020 were primarily driven by the original Wuhan (WA1/2020) strain [24,25,26]. These early cases occurred prior to the widespread availability of antivirals, monoclonal antibodies, and vaccines—interventions that likely modulate the risk of PNS/CNS involvement [27]. As such, this cohort offers the most robust longitudinal dataset for understanding the long-term neurological consequences of SARS-CoV-2 infection in the absence of mitigating therapies and the rise in viral variants.

## 2. Materials and Methods

### 2.1. Viruses and Cells

SARS-CoV-2 USA-WA1/2020 (NR-52281; BEI Resources; Manassas, VA, USA) was propagated using Vero E6 cells (CRL-1586, ATCC; Manassas, VA, USA) as previously described [28]. Viral stocks were titrated in duplicate via a standard agarose overlay plaque assay using the same cells. Standard cell culture protocols were used to maintain cells. The genome of our newly propagated viral stock was sequenced by the VT-Molecular Diagnostics Laboratory (Fralin Biomedical Research Institute) and deposited in GenBank (Accession# OP934235.1).

### 2.2. Guinea Pig Infections, Assessment, Tissue Collection and Processing

After a two-day acclimation period, three-week-old immunocompetent female Dunkin-Hartley guinea pigs (Charles River Laboratories, Wilmington, MA, USA) were infected, under ketamine/xylazine anesthesia, with either 10^3^ PFU (*n* = 12) or 10^5^ PFU (*n* = 12) of SARS-CoV-2 USA-WA1/2020 diluted in 1× PBS in our on-campus AAALAC-accredited ABSL-3 facility. The 20 µL inoculum was split between the nares for each guinea pig (10 µL in each naris). The 20 µL inoculum size was selected to minimize potential adverse effects from larger volumes, which could overwhelm the nasopharyngeal mucosa and lead to non-physiological viral distribution through aspiration and introduction of liquid into the lungs. This volume also aligns with the inoculum used in similar SARS-CoV-2 neuroinvasion studies in our lab with K18-hACE2 mice, wild-type C57BL/6J mice, and golden Syrian hamsters, allowing for direct comparison across these models [28]. Aliquots of the inocula and viral stock were saved for back titration using plaque assay to confirm infectious viral titer. Upon back titration, the viral stock was found to be 10^7^ PFU/mL and the two inocula were 10^5^ PFU and 10^3^ PFU in 20 uL. Uninfected controls (*n* = 2) were housed in our on-campus AAALAC-accredited ABSL-2 facility. Guinea pigs were monitored daily for weight, temperature, signs of distress, development of neurological complications, and survival. Weight (grams) for each animal was recorded daily and reported as the mean percentage increase/decrease for each inoculum group relative to the mean starting weight for that group. Temperature (°C) for each animal was recorded daily and reported as the mean percentage increase/decrease for each inoculum group relative to the mean pre-infection temperature for that group. Guinea pigs (*n* = 6 per group, per timepoint) from each inoculum group (10^5^ PFU, 10^3^ PFU) were euthanized at 3 days post infection (dpi) and 6 dpi. Tissues collected included CNS tissues (olfactory bulb, cerebellum, brainstem, cortex, hippocampus), PNS tissues (autonomic superior cervical ganglia-SCG; sensory trigeminal ganglia-TG and sensory dorsal root ganglia-DRG), lung, and whole blood. Half of the tissues from all animals were collected in TRI Reagent for RNA extraction and quantification of viral RNA copy number via RT-qPCR. The other half of the tissues were either flash frozen for detection of infectious virus via plaque assay (four of six animals per inoculum group per timepoint) or were collected in 10% formalin for detection of viral nucleocapsid via immunostaining (two of six animals per inoculum group per timepoint). When processing brains, the brains were split sagittally, maintaining attachment with the olfactory bulb. One hemisphere was fixed in formalin for immunostaining and the other dissected out into individual brain regions with each placed in TRI Reagent for RT-qPCR or flash frozen for plaque assays.

### 2.3. RNA Extraction and SARS-CoV-2 RT-qPCR

RNA was extracted using a standard guanidinium thiocyanate-phenol-chloroform extraction as previously described [28]. RNA purity and quantity were assessed using a NanoDrop 2000 spectrophotometer (ThermoFisher, Waltham, MA, USA). SARS-CoV-2 RT-qPCR reactions (10 µL) using the iTaq Universal Probe One-Step Kit (BioRad, Hercules, CA, USA) and SARS-CoV-2 N1 primers/probe mix (Integrated DNA Technologies, Coralville, IA USA) were performed on a ViiA 7 Real-Time PCR system (ThermoFisher, Waltham, MA, USA) as described in the instructions for use of the CDC 2019-Novel Coronavirus (2019-nCoV) Real-Time RT-PCR assay. Cycle conditions were as follows: Standard setting; 50 °C (10 min, 1 cycle), 95 °C (2 min, 1 cycle), followed by 95 °C (30 s) and 55 °C (3 s) for 45 cycles. Results were reported as virus RNA copy numbers per 200 ng total RNA.

### 2.4. Plaque Assays

To confirm the concentration of infectious virus used in the inocula and to assess for infectious virus in lung homogenate, undiluted and ten-fold serial dilutions of inocula or lung homogenate, in duplicate, were incubated on monolayers of Vero E6 cells in 24-well plates as previously described [28]. After 1 h of adsorption, the inoculum was removed, a 0.5% agarose overlay added, and plates were incubated for 48 h at 37 °C with 5% CO_2_. Plates were then fixed with 10% formaldehyde, the agarose overlay was removed, and cells were stained with plaque dye. For tissue homogenates, infectious viral titer was calculated as plaque-forming units per mg of homogenate (PFU/mg). For viral inoculum, infectious viral titer was calculated as PFU per mL (PFU/mL).

### 2.5. Immunostaining

Tissues were prepared for immunostaining as previously described [28]. Briefly, tissues were fixed in 10% formalin overnight, moved to 30% sucrose overnight, and subsequently embedded in optimal cutting temperature (OCT) media and frozen (ThermoFisher, Waltham, MA, USA). A Leica CM3050-S cryostat (Leica Biosystems, Danvers, MA, USA) was used to prepare 7 µm sections from each tissue block. For immunostaining, slides were rinsed in 1× PBS then blocked in 3% normal rabbit serum, 0.1% Triton X-100, and 1× PBS for 30 min at room temperature. SARS-CoV-2 nucleocapsid was visualized using an Alexa Fluor^®^ 488 conjugated rabbit monoclonal anti-SARS-CoV-2 nucleocapsid antibody at a 1:1000 concentration (NBP2-90988AF488; Novus Biologicals, Centennial, CO, USA). Neurons were visualized using an Alexa Fluor^®^ 647 conjugated rabbit monoclonal anti-NeuN antibody at a 1:1000 concentration (ab190565; Abcam, Waltham, MA, USA). Nuclei were visualized with 4′,6-diamidino-2-phenylindole (DAPI) in SlowFade Diamond antifade mounting medium (ThermoFisher, Waltham, MA, USA). Primary antibodies were incubated overnight at 4 °C in 1% normal rabbit serum, 0.1% Triton-100X, and 1× PBS. All antibodies used were previously validated in our lab via immunostaining and Western blot [28]. Lungs were immunostained and imaged as these are a primary site of infection of SARS-CoV-2 and would serve as a confirmation of the ability of guinea pigs to be infected by the virus. Brains were immunostained and imaged as confirmation of neuroinvasive infection. The olfactory bulb, cortex, and midbrain were imaged as these areas are commonly infected, as has been reported in studies using mice and hamsters [16,29,30,31]. The cerebellum was not imaged as this area is largely spared substantial infection even in highly susceptible animal models [28,32]. To confirm the specificity of the SARS-CoV-2 nucleocapsid antibody, we stained positive control tissues (lung and brain) from K18-hACE2 mice that were intranasally infected with 10^5^ PFU of SARS-CoV-2 WA1/2020 and euthanized at 6 dpi as part of a separate neuroinvasion study in our lab, as previously described [28]. Imaging was performed using a Leica SP8 scanning confocal microscope (Leica Biosystems, Danvers, MA, USA) with common imaging settings across specimens of the same tissue type. Images were composed using ImageJ (version 1.54).

### 2.6. Quantification and Statistical Analysis

RT-qPCR results that fell below the lower limit for the standard curve (80 copies) after normalization were reported as zero for inclusion in the analysis, no data were excluded. All statistical analyses were performed in GraphPad Prism version 8 (Dotmatics, Boston, MA, USA). For statistical analysis, significance was set at *p* < 0.05 and calculated using a mixed-effects analysis. If significance was found, pairwise analysis was performed using Tukey’s honestly significant difference (HSD) post hoc test to identify the comparison(s) with significant difference(s).

## 3. Results

### 3.1. Clinical Assessment Revealed No Overt Disease Following SARS-CoV-2 Inoculation

To determine the suitability of immunocompetent wild-type Dunkin-Hartley guinea pigs in the assessment of ancestral SARS-CoV-2 infection of the PNS and CNS, we intranasally inoculated guinea pigs with 10^3^ PFU or 10^5^ PFU of SARS-CoV-2 USA-WA1/2020, monitored them daily (weight, temperature, distress, neurological complications, survival), and collected tissues at three- and six-days post infection (dpi) (Figure 1A). Weight loss following infection was not observed in either inoculum group and both groups continued to gain weight, with ≈20% mean weight gain in each inoculum group by 6 dpi. While infection did not cause weight loss, weight gain in both inoculum groups was slower over time when compared with uninfected controls. The rate of weight gain between both inoculum groups and controls diverged at 1 dpi and remained so up to 6 dpi, with the control group showing a ≈40% mean weight gain by 6 dpi (Figure 1B). A statistically significant, though not clinically significant, difference in weight gain between the 3 log PFU vs. 5 log PFU group (*p* = 0.005) and 3 log PFU vs. uninfected group (*p* = 0.025) was detected at 3 dpi. While this difference reached statistical significance, it was likely not biologically significant as no animal in either group displayed signs of disease or distress. Similarly, a statistically significant, though not clinically significant, difference was also detected between the 3 log PFU vs. uninfected groups at 4 dpi (*p* = 0.026). The differences in the rate of weight gain were not clinically significant. A transient increase in temperature (≈1 °C) was noted in the 3 log PFU inoculum group at 3 dpi, which was resolved by 4 dpi (Figure 1C). An increase in temperature (≈2 °C) was noted in the 5 log PFU inoculum group beginning 1 dpi and lasting to 6 dpi. A statistically significant, though not clinically significant, difference was detected in temperatures between the 3 log PFU vs. 5 log PFU groups at 1 dpi: (*p* = 0.027). (Figure 1C). No signs of distress or neurological involvement were observed in guinea pigs from either inoculum group, and all animals survived to their predetermined endpoints (Figure 1D).

### 3.2. SARS-CoV-2 RNA and Nucleocapsid Are Undetectable in Tissues Three and Six Days Post Infection

The presence of SARS-CoV-2 RNA in CNS tissues (olfactory bulb, hippocampus, brainstem, cerebellum, cortex), PNS tissues (autonomic superior cervical ganglia-SCG; sensory dorsal root ganglia-DRG and trigeminal ganglia-TG), lungs, and whole blood was assessed by RT-qPCR. Viral RNA was not detected in any CNS tissue, PNS tissue, or lung at either timepoint in either inoculum group. A low-level viremia (113 RNA copies) was detected in one of the six guinea pigs in the 5 log PFU inoculum group at 3 dpi; however, all other tissues from this guinea pig and all tissues from all other guinea pigs were negative for SARS-CoV-2 RNA (Figure 1E). No infectious virus was detected in lung homogenates by plaque assay at either 3 or 6 dpi in either inoculum group. Immunostaining for SARS-CoV-2 nucleocapsid (regions imaged are illustrated in Figure 2A) in the olfactory bulb (Figure 2B), cortex (Figure 2C), midbrain (Figure 2D), and lungs (Figure 3) did not detect SARS-CoV-2 nucleocapsid in these tissues at 6 dpi. Due to the absence of viral RNA in most tissues, viral protein in the lungs or brain, and infectious virus in the lung—the primary site of SARS-CoV-2 replication—additional plaque assays and immunostaining were not conducted.

## 4. Discussion

During the 2002–2004 SARS-CoV-1 epidemic, guinea pigs were investigated as a potential small animal model for evaluating post-infection pathology and vaccine safety. While infected guinea pigs did not exhibit overt disease, they did develop transient temperature increases, exhibited slowed weight gain, and developed both pulmonary and extrapulmonary pathologies akin to those observed in humans. Guinea pigs infected with SARS-CoV-1 also produced infectious virus recoverable from the lungs and generated virus-specific neutralizing antibody responses [8,9]. Following the creation of mouse-adapted SARS-CoV-1 viruses, research shifted towards employing mouse models for pathology studies, while guinea pigs were used for immunological studies [33,34,35]. While the use of mouse-adapted viruses allowed for rapid study of host/virus interactions, they were limited due to their non-human tropism, potentially harboring mutations in viral components crucial for human disease [36]. To address these constraints, several years after the end of the SARS-CoV-1 epidemic, transgenic mice expressing human ACE2 (hACE2), the receptor used for attachment and entry by the virus, were developed. Infection in these mice is marked by viral replication in respiratory epithelial cells, viral dissemination to extrapulmonary tissues, and rapid death following neuroinvasion of the central nervous system [10].

Upon confirmation that SARS-CoV-2 utilized the same receptor as SARS-CoV-1, these hACE2 transgenic mice provided a rapidly accessible small animal model for SARS-CoV-2 pathology research [37,38,39]. As the SARS-CoV-2 pandemic progressed, reports of neurological complications affecting both the CNS and PNS became increasingly common [21,22,23,40]. To investigate neuroinvasion of the CNS, several groups intranasally inoculated hACE2 mice with SARS-CoV-2. These studies showed that the virus is capable of infecting sustentacular cells in the nasal epithelium and to a lesser extent, olfactory sensory neurons, thus gaining entry into the CNS through these cells. These hACE2 mice succumbed to infection rapidly following neuroinvasion, mirroring findings from previous studies involving SARS-CoV-1 in these mice [16,19,29]. Such robust neuroinvasion and subsequent rapid mortality are not often reported in humans. Therefore, the comparatively milder disease exhibited in guinea pigs compared to the severe disease observed in hACE2 mice when inoculated with SARS-CoV-1 makes guinea pigs an attractive alternative model to study neuroinvasion of coronaviruses.

Several in silico and in vitro studies have suggested that guinea pigs may be resistant to infection with human coronaviruses [41]. Human embryonic kidney cells (AD293 cells) transfected to express guinea pig ACE2 (gpACE2) are not rendered susceptible to infection with SARS-CoV-1, likely owing to the low sequence homology (86.8%) between gpACE2 with hACE2, low expression post transfection (≈20% cells), and overall low infection rate (≈20% cells) [42]. Similar low gpACE2 expression was detected in transfected BHK-21 cells used to assess interactions with SARS-CoV-2, leading to the suggestion that available sequences for gpACE2 may be incomplete or that guinea pigs lack functional ACE2 [41]. An in silico analysis of interactions between gpACE2 and the receptor binding domain (RBD) of SARS-CoV-2 spike reveals that gpACE2 has a reduced affinity for the RBD compared to hACE2, which is attributed to differences in the number and types of contacts between the RBD and ACE2. The reduced affinity of gpACE2 for SARS-CoV-2 spike RBD may provide a mechanism to explain the previous in vitro findings, while also raising doubts about the suitability of guinea pigs in immunological studies for the development of SARS-CoV-2/ACE2 targeting antibodies/therapeutics [43]. Validation of these predictions from in silico and in vitro studies regarding the infectability of the peripheral nervous system of guinea pigs with SARS-CoV-2 has yet to be reported in vivo.

Our detection of non-clinically significant mild temperature fluctuations and slowed weight gain in otherwise asymptomatic guinea pigs inoculated with SARS-CoV-2 mirrors findings from previous studies using SARS-CoV-1. These transient changes in temperature and weight gain may signify a brief innate immune response responsible for clearing the virus prior to the establishment of infection. Unlike prior SARS-CoV-1 studies, in which viral RNA and infectious virus were detected up to 18 dpi, we found no detectable viral RNA or infectious virus in the lungs of SARS-CoV-2-inoculated guinea pigs [8,9]. This lack of recoverable virus is likely attributed to the route of infection used and the previously discussed suboptimal binding kinetics between gpACE2 and the SARS-CoV-2 spike protein. Previous studies assessing SARS-CoV-1 infection in guinea pigs relied on intraperitoneal injection (IP) for virus inoculation [36]. Intraperitoneal injection of a respiratory virus is not physiologically relevant to human disease, as it could lead to widespread dissemination of the virus, bypassing the restrictions imposed by gpACE2-mediated infection. Our use of intranasal inoculation, in contrast to IP inoculation, likely contributes to differences between viral dissemination of SARS-CoV-1 in previous studies and our study with SARS-CoV-2.

Recently, another research group confirmed that guinea pigs are resistant to intranasal infection with both ancestral SARS-CoV-2 and more contemporary variants [44]. Notably, Iwatsuki-Horimoto et al. used an ancestral SARS-CoV-2 strain carrying the D614G spike mutation (SARS-CoV-2/UT-HP095-1N/Human/2020/Tokyo), a variant associated with increased infectivity, whereas we used hCoV-19/USA-WA1/2020, an early ancestral isolate that lacks the D614G mutation and retains a threonine at position 614, thus more closely resembling the original Wuhan-like virus. Taken together, these findings indicate that the D614G mutation alone is not sufficient to enable generalized infection of guinea pigs (nasal turbinates, lungs) or to support neuroinvasion of PNS innervations of the oronasal mucosa and their functionally connected CNS regions. Additionally, we previously demonstrated in C57BL/6J mice that neuroinvasion can occur in the absence of physical signs [28]. Specifically, we identified viral RNA (ranging from 1 to 5 log copies per 200 ng of total RNA), viral nucleocapsid protein (via immunostaining), and infectious virus (3–5 PFU/mg of tissue homogenate) within the trigeminal ganglia, superior cervical ganglia, dorsal root ganglia, spinal cord, and functionally connected brain regions despite the absence of weight loss or neurological signs. These findings underscore the importance of directly examining nervous system tissues, even when clinical disease is not apparent, to accurately assess neuroinvasion. Our current study complements the work of Iwatsuki-Horimoto et al., who focused on viral detection in nasal turbinates and lungs, by emphasizing the critical need to evaluate PNS and CNS tissues for a comprehensive understanding of the neuroinvasive potential of SARS-CoV-2. Finally, neither of our studies detected clinical signs of infection, impaired weight gain, viral RNA, or infectious virus in the lungs of SARS-CoV-2 inoculated guinea pigs, which demonstrates the replicability of our findings.

While our research indicates that guinea pigs are not suitable for investigating neuroinvasion, they have continued to be utilized during the SARS-CoV-2 pandemic as models for studying SARS-CoV-2 immunological responses. Guinea pigs have been used to evaluate vaccine safety, tolerability, immunogenicity, and neutralization efficacy against various SARS-CoV-2 variants [45,46,47,48,49,50,51,52,53,54,55,56,57,58,59]. Additionally, they have been used to investigate the sustained effectiveness of cross-protection conferred by antibodies resulting from exposure to ancestral SARS-CoV-2 and other variants of interest, including those harboring significant RBD mutations, as well as seasonal coronaviruses [60,61,62,63,64]. Guinea pigs have also been used to develop rapid antigen tests for SARS-CoV-2 [65,66,67].

Of note, to assess the potential for secondary spillover, guinea pigs were included in the epidemiological investigation of two case reports involving owners/handers with symptomatic RT-qPCR-positive SARS-CoV-2 infections. Of a menagerie of SARS-CoV-2 exposed animals, including hamsters, mice, rabbits, chinchillas, and guinea pigs, tested via RT-qPCR, only hamsters were found to have contracted SARS-CoV-2 infection after exposure to a symptomatic COVID-19 case [68,69]. These findings are not surprising, as hamsters are a widely used animal model for SARS-CoV-2 infection [70,71,72,73,74]. The results of these epidemiological investigations provide further support that guinea pigs are not susceptible to infection when exposed to SARS-CoV-2 via a respiratory infection route.

## 5. Conclusions

Through the use of multiple complementary assays for the detection of viral RNA, viral protein, and infectious virus, we show that immunocompetent female Dunkin-Hartley guinea pigs are not susceptible to intranasal infection with ancestral SARS-CoV-2 isolate USA-WA1. Our findings provide direct in vivo confirmation of predictions proposed by previous in silico and in vitro studies calling into question the ability of SARS-CoV-2 to infect guinea pigs.

## Figures and Tables

**Figure 1 viruses-17-00706-f001:**
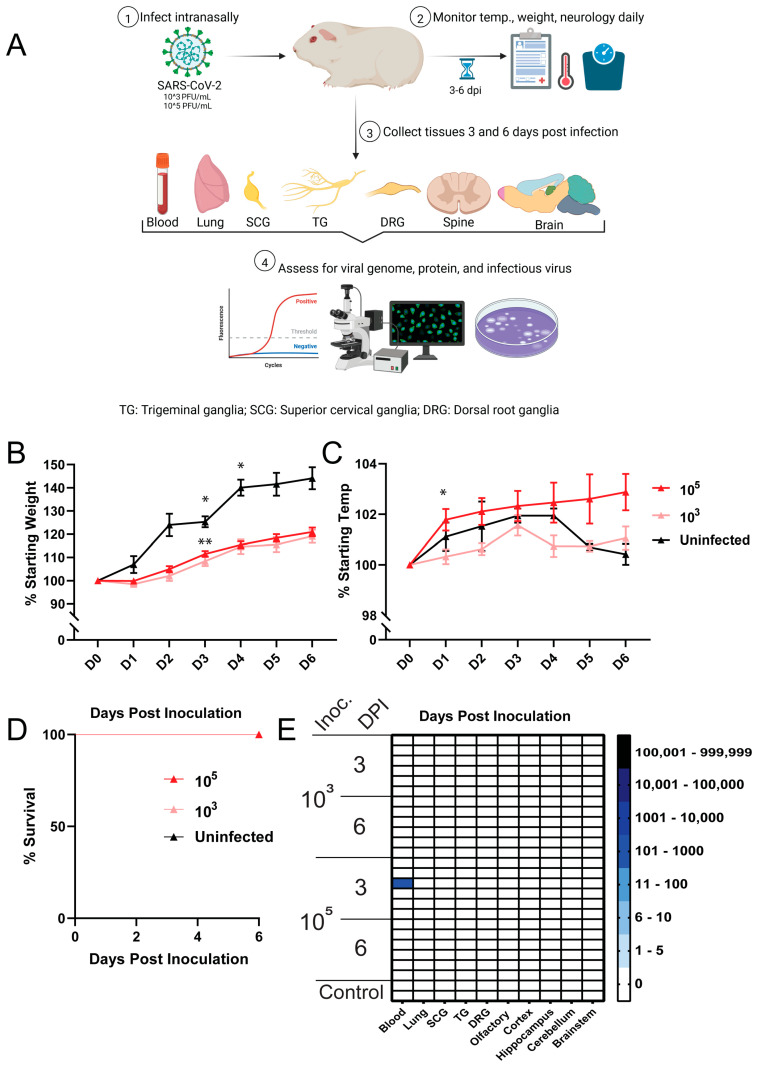
Clinical assessment of guinea pigs. (**A**) Experimental design showing that guinea pigs were inoculated with either 10^3^ PFU (*n* = 12) or 10^5^ PFU (*n* = 12) of SARS-CoV-2 USA-WA1/2020 and monitored daily for weight, temperature, signs of distress, development of neurological complications, and survival before tissue collection at 3 and 6 dpi (*n* = 6 from each group per timepoint). (**B**) Post-infection weight loss did not occur, though weight gain was slower at 3 dpi in the 3 log PFU vs. 5 log PFU groups (*p* = 0.005) and 3 log PFU vs. uninfected groups (*p* = 0.025), as well as the 3 log PFU vs. uninfected groups at 4 dpi (*p* = 0.026); (**C**) Clinically significant fevers did not develop post-inoculation, however a transient increase in temperature was detected between the 3 log PFU vs. 5 log PFU groups at 1 dpi: (*p* = 0.027); (**D**) All animals survived to their predetermined timepoints; (**E**) Heatmap of RT-qPCR assay results showing only one guinea pig from the 5 log PFU group had a low-level transient viremia (113 RNA copies) at 3 dpi, all other tissues from this guinea pig and all tissues from all other guinea pigs were negative for SARS-CoV-2 RNA. * *p* < 0.05, ** *p* < 0.01.

**Figure 2 viruses-17-00706-f002:**
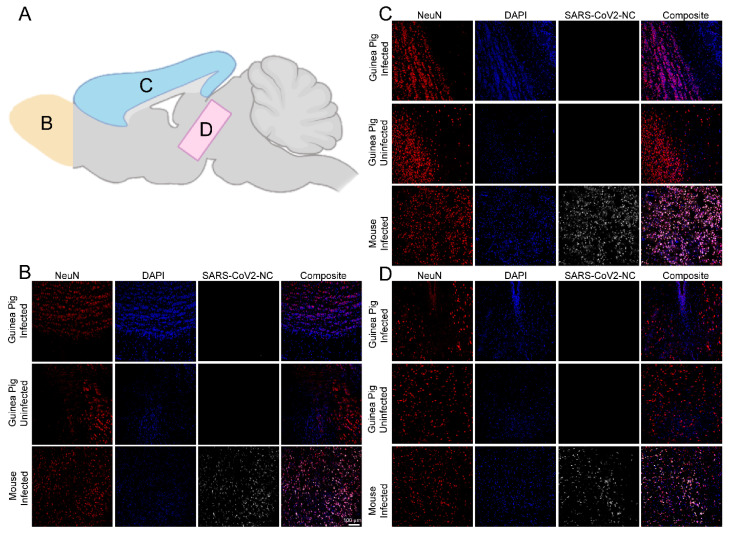
Immunofluorescent imaging of olfactory bulb, cortex, and midbrain. Sagittal sections of brains from 5 log PFU-infected guinea pigs, uninfected guinea pigs, and 5 log PFU-infected k18-hACE2 mice (positive control) were stained for the presence of SARS-CoV-2 nucleocapsid (SARS-CoV-2-NC, white), neurons (NeuN, red), and DNA (DAPI, blue). Images from infected animals are from 6 dpi. (**A**) Diagram of areas of the brain included in imaging. (**B**) Infection throughout the olfactory bulb is present in positive control mice and absent in infected and uninfected guinea pigs; (**C**) Infection throughout the cortex is present in positive control mice and absent in infected and uninfected guinea pigs; (**D**) Infection throughout the midbrain is present in positive control mice and absent in infected and uninfected guinea pigs.

**Figure 3 viruses-17-00706-f003:**
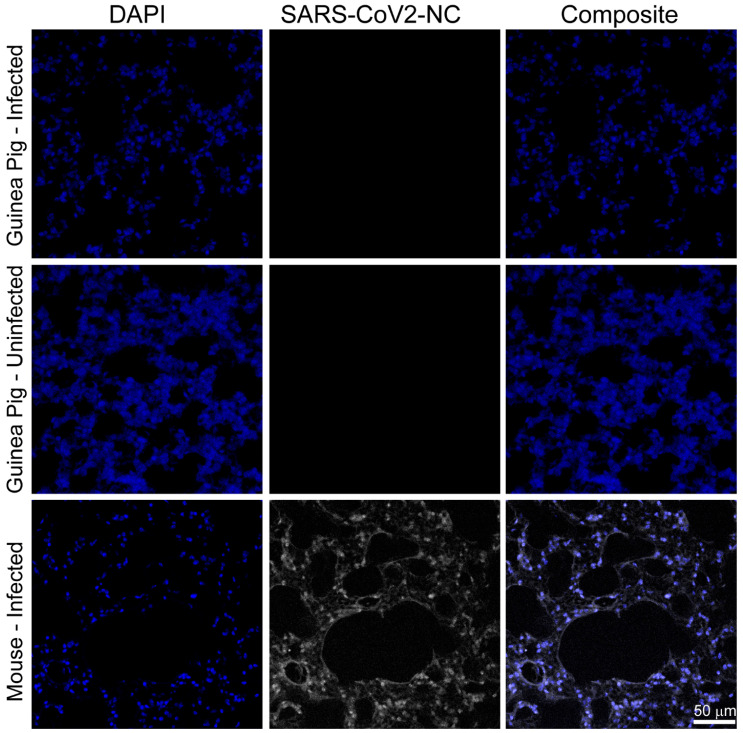
Immunofluorescent imaging lungs. Lungs were sectioned and stained for SARS-CoV-2 nucleocapsid (SARS-CoV-2-NC, white) and DNA (DAPI, blue) from guinea pigs infected with SARS-CoV-2, uninfected guinea pigs, and k18-hACE2 mice infected with SARS-CoV-2. Images from infected animals are from 6 dpi. Infection throughout the lung is present in positive control mice and absent in infected and uninfected guinea pigs.

## Data Availability

The raw data supporting the conclusions of this article will be made available by the authors upon reasonable request made to the corresponding author.

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
