# Peer review of "Guinea Pigs Are Not a Suitable Model to Study Neurological Impacts of Ancestral SARS-CoV-2 Intranasal Infection"

_viruses, 2025, doi:10.3390/v17050706_

Round 1
Reviewer 1 Report
Comments and Suggestions for Authors
- There had reports that guinea pigs are not suitable models for SARS-CoV-2 infection as the following papers. Why the authors did this study?
https://www.nature.com/articles/s44298-024-00068-8
https://onlinelibrary.wiley.com/doi/10.1111/xen.12772
- Did the author check SARS-CoV-2 infection in pig cell line before vivo experiment?
- In the introduction line 57, the author should describe the hACE-2 K-18 mice used as vivo model in SARS-CoV-2 infection.
- Did the author try SARS-COV-2 strain adapted in Guinea pig and tried for infection? For example, BALB/C mice are not vivo model for SARS-CoV-2 infection. But researchers tried BALB/C mouse adapted strain in the following paper. https://pubmed.ncbi.nlm.nih.gov/36728430/
- Please discuss that guinea pigs gave different response between SRS-CoV-1 and SARS-CoV-2 infection?
Author Response
Comment 1: There had reports that guinea pigs are not suitable models for SARS-CoV-2 infection as the following papers. Why the authors did this study?
Response 1: We thank the reviewer for bringing these papers to our attention and will address each individually.
- Sensitivity of rodents to SARS-CoV-2: Gerbils are susceptible to SARS-CoV-2, but guinea pigs are not (https://www.nature.com/articles/s44298-024-00068-8)
- Context of Work: We appreciate the reviewer’s observation and agree studies similar to ours, including work by Iwatsuki-Horimoto et al., have been published, however, we would like to clarify the timeline of our own study to provide proper context. The research reported in our manuscript originated during the early stages of the COVID-19 pandemic and was first shared as a preprint on bioRxiv on 20 May 2022 (link below), preceding Iwatsuki-Horimoto et al. (21 November 2024). In this manuscript, our lab investigated the susceptibility of the peripheral nervous system (PNS) to SARS-CoV-2 infection using both K18-hACE2 transgenic and wild-type C57BL/6J mice, as well as primary neuronal cultures derived from both. Based on our findings, we sought to validate PNS infection in an alternative animal model. At that time, golden Syrian hamsters had not yet become the standard, and we initially selected guinea pigs due to their known susceptibility to SARS-CoV-1 and our lab's extensive experience with this species. During peer review of our initial manuscript, reviewers suggested replacing the negative guinea pig data with data from golden Syrian hamsters, which by then had become the preferred model. We complied with this recommendation, removed our negative data from guinea pigs, confirmed PNS infection in hamsters, and subsequently published our findings (link below). We hope this clarifies the provenance of our study and its distinction in timing and approach.
- bioRxiv: https://www.biorxiv.org/content/10.1101/2022.05.20.492834v1.full
- Manuscript: https://pmc.ncbi.nlm.nih.gov/articles/PMC11311394/
- Differences to other studies: 1)Viruses: While Iwatsuki-Horimoto et al. conducted a study similar in scope to ours, there are important distinctions worth highlighting. In particular, the viral isolates used in each study differ in their harboring the D614G mutation. Iwatsuki-Horimoto et al. used an ancestral SARS-CoV-2 strain carrying the D614G spike mutation (SARS-CoV-2/UT-HP095-1N/Human/2020/Tokyo), a variant associated with increased infectivity. In contrast, we used hCoV-19/USA-WA1/2020, an ancestral isolate that lacks the D614G spike mutation (retains a threonine at position 614), and is therefore closer to the original Wuhan-like isolate. By combining our work and that of Iwatsuki-Horimoto et al. readers can directly assess how this single amino acid change influences tissue tropism, as it is not sufficient for generalized infection (nasal turbinates, lungs) or neuroinvasion of PNS innervations of the oronasal mucosa and functionally connected CNS areas. 2)Anatomical sites assessed: The scope of tissue analysis in our study differs from that of Iwatsuki-Horimoto et al., which focused primarily on viral infection in the nasal turbinates and lungs. In contrast, our investigation aimed to determine whether SARS-CoV-2 can invade the PNS and CNS. As we have previously shown in C57BL/6J mice, neuroinvasion can occur in the absence of overt clinical symptoms. Consistent with these findings, we observed no weight loss or neurological signs in infected mice, yet detected clear evidence of neuroinvasion. Specifically, we identified viral RNA (ranging from 1 to 5 log copies per 200 ng of total RNA), nucleocapsid (via immunostaining), and infectious virus (3–5 PFU/mg of tissue homogenate) in the trigeminal ganglia, superior cervical ganglia, dorsal root ganglia, spinal cord, and functionally connected brain regions. These findings underscore the importance of directly assessing nervous system tissues, even in the absence of clinical disease, to rule out neuroinvasion. Our study includes such data for guinea pigs—data not provided by Iwatsuki-Horimoto et al.—thereby offering a more comprehensive evaluation of SARS-CoV-2’s neuroinvasive potential.
- 3)Mechanism to explain success of SARS-CoV-1 infection when guinea pigs are clearly resistant to CoVs: We offer an explanation for the discrepancy between earlier reports of successful SARS-CoV-1 infection in guinea pigs and the lack of detectable infection in our study and that of Iwatsuki-Horimoto et al. A key distinction lies in the route of viral inoculation. Previous studies that detected infection in guinea pigs used intraperitoneal inoculation, whereas both our study and Iwatsuki-Horimoto et al. used intranasal inoculation. This difference in inoculation route significantly influences viral dissemination and tissue tropism, as intraperitoneal administration bypasses the mucosal barrier of ACE2-expressing respiratory epithelial cells. Instead, it facilitates hematogenous spread via lymphatic drainage to regional lymph nodes and subsequently into systemic circulation.
- We have updated our discussion section to include Iwatsuki-Horimoto et al.
- 4) Reproducibility: Finally, our work supports that of Iwatsuki-Horimoto et al. as neither of our studies detected clinical signs of infection, impaired weight gain, viral RNA, or infectious virus in the lungs of SARS-CoV-2 inoculated guinea pigs, which demonstrates that these findings are replicable by different research groups.
- SARS-CoV-2 does not infect pigs, but this has to be verified regularly (https://onlinelibrary.wiley.com/doi/10.1111/xen.12772)
- This article is a review summarizing what is known about the susceptibility of pigs to infection by SARS-CoV-2, with a focus on implications for xenotransplantation. While the review briefly mentions guinea pigs' resistance to SARS-CoV-2 infection based on an in silico modeling study suggesting poor ACE2–spike binding (DOI: 10.15212/ZOONOSES-2021-0010), our manuscript also discusses this point and adds additional in vitro studies as well (PMID: 33347434). Both the review and our manuscript cite a study demonstrating guinea pigs' susceptibility to SARS-CoV-1 infection (PMID: 16101345); however, we cite an additional study, confirming SARS-CoV-1 infection, not included in the review (PMID: 15508583). Furthermore, our manuscript provides a mechanistic explanation for the differing susceptibilities of guinea pigs to SARS-CoV-1 versus SARS-CoV-2. Specifically, SARS-CoV-1 infection was established using intraperitoneal inoculation, which bypasses mucosal ACE2-mediated entry and facilitates hematogenous viral spread, whereas our use of intranasal inoculation is more physiologically relevant and reflects natural ACE2 binding barriers to infection. Given these considerations, we do not believe this review article meaningfully adds to the discussion of our manuscript.
Comment 2: Did the author check SARS-CoV-2 infection in pig cell line before vivo experiment?
Response 2: We did not perform in vitro infections using guinea pig cell lines because available lines are limited to fibroblasts (lung, fetal), epithelial cells (colon), and lymphocytes (B cells). No established cell lines derived from the PNS or CNS tissues—our primary tissues of interest—are currently available, and thus the existing cell lines are not suitable for addressing the specific neurotropism we are investigating.
Comment 3: In the introduction line 57, the author should describe the hACE-2 K-18 mice used as vivo model in SARS-CoV-2 infection.
Response 3: The K18-hACE2 transgenic mouse model, developed by Stanley Perlman following the SARS-CoV-1 epidemic, expresses hACE2 receptor under the control of the human cytokeratin 18 (K18) promoter (PMID: 17079315). This promoter drives expression primarily in epithelial tissues, including the respiratory tract, a primary target for SARS-CoV-2 infection. These mice are highly susceptible to SARS-CoV-2, developing robust viral replication in the lungs and, in severe cases, viral dissemination to other organs such as the brain. The K18-hACE2 model has been widely used to study SARS-CoV-2 pathogenesis, host responses, and therapeutic interventions in vivo, however the severe disease exhibited during infection made them an inferior model to golden Syrian hamsters (PMID: 33257679, 33166988, 33073694, 32839612, 32841215). We have updated our introduction to include this justification.
Comment 4: Did the author try SARS-COV-2 strain adapted in Guinea pig and tried for infection? For example, BALB/C mice are not vivo model for SARS-CoV-2 infection. But researchers tried BALB/C mouse adapted strain in the following paper. https://pubmed.ncbi.nlm.nih.gov/36728430/
Response 4: We did not attempt to generate a guinea pig-adapted SARS-CoV-2 strain, as has been done in other studies with mouse-adapted variants. We intentionally avoided the use of rodent-adapted strains, as these can acquire mutations affection viral proteins that play roles in pathogenesis in humans. While a guinea pig-adapted SARS-CoV-2 strain may enhance infectivity or neuroinvasion in rodents, the resulting infection may not accurately reflect the mechanisms of SARS-CoV-2 infection and disease progression in humans, potentially limiting the translational relevance of our findings.
Comment 5: Please discuss that guinea pigs gave different response between SRS-CoV-1 and SARS-CoV-2 infection?
Response 5: We have discussed the different response of guinea pigs to SRS-CoV-1 and SARS-CoV-2 infection in multiple parts of our manuscript. We note this the: 1) Introduction: “During the SARS-CoV-1 epidemic of 2002-2004, early research into the virus showed that guinea pigs were susceptible to infection and developed clinical signs and pulmonary pathologies similar to those observed in humans”, and 2) Discussion “During the 2002-2004 SARS-CoV-1 epidemic, guinea pigs were investigated as a potential small animal model for evaluating post-infection pathology and vaccine safety. While infected guinea pigs did not exhibit overt disease, they did develop transient temperature increases, exhibited slowed weight gain, developed both pulmonary and extrapulmonary pathologies akin to those observed in humans. Guinea pigs infected with SARS-CoV-1 also produced infectious virus recoverable from the lungs and generated virus specific neutralizing antibody responses”, “Our detection of non-clinically significant mild temperature fluctuations and slowed weight gain in otherwise asymptomatic guinea pigs inoculated with SAS-CoV-2 mirrors findings from previous studies using SARS-CoV-1.”

Reviewer 2 Report
Comments and Suggestions for Authors
The manuscript entitled, “Guinea pigs are not a suitable model to study neurological impacts of ancestral SARS-CoV-2 intranasal infection,” is a short report documenting the authors’ efforts to establish the guinea pig model of neurological fallout from SARS-CoV-2 infection. Much like other recent efforts, the authors document the non-responsiveness of the guinea pig to SARS-CoV-2. The language is straightforward and mostly without error. The methods, results and discussion are also very straight forward. There are only two issues that the authors do not fully address that those are of the relevance to the literature and to establish that the animals were indeed infected. Perhaps the relevance to the literature might help interpret the lack of proof of infection.
- Line 61: It seems there’s a missing “%” after “>80.”
- The authors use the ancestral SARS-CoV-2 isolate, USA-WA1/2020; however, there is no timeline of the emergence of this variant or the associated neurovirulence. There should be some introduction or justification of the use of this variant’s use as relevant for neurological disease in context of the human cases, especially considering the short time that the ancestral variant was circulating.
- The authors chose to use a total volume of 20µL split between nares per guinea pig. This would mean that only 10µL was used on each side, a volume less than an order of magnitude from other intranasal guinea pig challenges. Please discuss the rationale for such a volume and its potential impact on the study. Specifically, the volume is low enough that there might be some doubt of infection; how would the authors argue against such doubt? Were the challenges, measurements, analyses, etc. performed in blinded fashion? I see that temperature is loosely used to suggest infection, but the uninfected animals were between the two different challenge groups, with error bars it’s hard to tell a difference; was there a confirmation through other means or timepoints beside the one animal on day 3? What can be said to counter the argument, “maybe they were not challenged?” What were the titers of the inoculum? Anything?
- Recently, 21 NOV 2024, a research group published a paper in npj Viruses entitled, “Sensitivity of rodents to SARS-CoV-2: Gerbils are susceptible to SARS-CoV-2, but guinea pigs are not,” in which the authors document the resistance to infection of Hartley guinea pigs exposed to different variants of SARS-CoV-2 intranasally. How are the findings of this manuscript different from those of the recently published or how do they support the authors’ findings? Please include reference to this manuscript and any other recent material when responding.
Author Response
Comment 1: Line 61: It seems there’s a missing “%” after “>80.”
Response 1: The reviewer is correct. We have corrected the error.
Comment 2: The authors use the ancestral SARS-CoV-2 isolate, USA-WA1/2020; however, there is no timeline of the emergence of this variant or the associated neurovirulence. There should be some introduction or justification of the use of this variant’s use as relevant for neurological disease in context of the human cases, especially considering the short time that the ancestral variant was circulating.
Response 2: We agree that investigating the neuroinvasive potential of more recent SARS-CoV-2 variants (e.g., Omicron) is important and will be a focus of our future research. However, our current study centers on the estimated 79.2 million COVID-19 cases from 2020, which were primarily driven by the original Wuhan (WA1/2020) strain (PMID: 34546094, 35602161, WHO). These early cases occurred prior to the widespread availability of antivirals, monoclonal antibodies, and vaccines—interventions that likely modulate the risk of PNS/CNS involvement (PMID: 34033342). As such, this cohort offers the most robust longitudinal dataset for understanding the long-term neurological consequences of SARS-CoV-2 infection in the absence of mitigating therapies and viral variants. We have updated our introduction to include this justification.
Comment 3: The authors chose to use a total volume of 20µL split between nares per guinea pig. This would mean that only 10µL was used on each side, a volume less than an order of magnitude from other intranasal guinea pig challenges. Please discuss the rationale for such a volume and its potential impact on the study. Specifically, the volume is low enough that there might be some doubt of infection; how would the authors argue against such doubt? Were the challenges, measurements, analyses, etc. performed in blinded fashion? I see that temperature is loosely used to suggest infection, but the uninfected animals were between the two different challenge groups, with error bars it’s hard to tell a difference; was there a confirmation through other means or timepoints beside the one animal on day 3? What can be said to counter the argument, “maybe they were not challenged?” What were the titers of the inoculum? Anything?
Response 3: The choice of a 20 µL inoculum was made based on several factors. While we acknowledge this volume is smaller than those typically used in guinea pig challenge studies, it was selected to minimize potential adverse effects from larger volumes, which could overwhelm the nasopharyngeal mucosa and lead to non-physiological viral distribution. Our goal was to concentrate infection in the nasopharynx, particularly in the peripheral nervous system innervations of the oronasal mucosa, rather than risk inducing deeper lung infection, which might occur with a larger inoculum. This volume also aligns with the inoculum used in similar SARS-CoV-2 neuroinvasion studies in our lab with K18-hACE2 mice, wild-type C57BL/6J mice, and golden Syrian hamsters, allowing for direct comparison across these models. Importantly, the inoculum was back-titrated to confirm the presence of infectious virus and ensure proper dosing. Upon back titration, the viral stock and two inocula were found to be 107 PFU/mL, 105 PFU, and 103 PFU, respectively. We have updated our methods section to include this justification.
Comment 4: Recently, 21 NOV 2024, a research group published a paper in npj Viruses entitled, “Sensitivity of rodents to SARS-CoV-2: Gerbils are susceptible to SARS-CoV-2, but guinea pigs are not,” in which the authors document the resistance to infection of Hartley guinea pigs exposed to different variants of SARS-CoV-2 intranasally. How are the findings of this manuscript different from those of the recently published or how do they support the authors’ findings? Please include reference to this manuscript and any other recent material when responding.
Response 4: We appreciate the reviewer’s observations and the opportunity to clarify the context of our study in relation to the recent publication by Iwatsuki-Horimoto et al. Our research, which began during the early stages of the COVID-19 pandemic, was first shared as a preprint on bioRxiv in May 2022 (link below), preceding the publication by Iwatsuki-Horimoto et al. (21 November 2024). Our study focused on the susceptibility of the peripheral nervous system (PNS) to SARS-CoV-2 infection using K18-hACE2 transgenic and wild-type C57BL/6J mice, as well as primary neuronal cultures from each. Initially, we selected guinea pigs as an alternative animal model due to their known susceptibility to SARS-CoV-1 and our lab’s extensive experience with this species. After peer review, we replaced the guinea pig data with golden Syrian hamster data, at the behest of a reviewer, confirming PNS infection and subsequently publishing our findings (link below). Additionally, while both studies used ancestral SARS-CoV-2 isolates, there is a key difference in the viral strains used: Iwatsuki-Horimoto et al. used a strain harboring the D614G mutation, whereas we employed the original Wuhan-like isolate (hCoV-19/USA-WA1/2020), which lacks this mutation. This allows for direct comparison of how this single amino acid change affects tissue tropism and neuroinvasion, with our data indicating that the mutation does not alter PNS/CNS infection. Furthermore, while Iwatsuki-Horimoto et al. focused on the nasal turbinates and lungs, we expanded our investigation to include PNS innervations of the oronasal mucosa and functionally connected CNS regions. We have previously demonstrated in C57BL/6J mice that neuroinvasion can occur without overt clinical symptoms, as evidenced by the detection of viral RNA, nucleocapsid, and infectious virus in the PNS of the oronasal mucosa and functionally connected CNS regions, despite the absence of weight loss or neurological signs in infected mice. To rule out neuroinvasion in an otherwise non-systemically infected animal model, nervous system tissues must be assessed. Additionally, we offer a possible explanation for the discrepancy between earlier reports of successful SARS-CoV-1 infection in guinea pigs and the lack of detectable infection in our and Iwatsuki-Horimoto et al. studies. A key distinction lies in the route of viral inoculation. Previous studies demonstrating infection in guinea pigs used intraperitoneal inoculation, whereas both our study and Iwatsuki-Horimoto et al. used intranasal inoculation. This difference in inoculation route significantly influences viral dissemination and tissue tropism, as intraperitoneal administration bypasses the mucosal barrier of ACE2-expressing respiratory epithelial cells. Instead, it facilitates hematogenous spread via lymphatic drainage to regional lymph nodes and subsequently into systemic circulation. We hope this clarifies the provenance of our work and illustrates how it complements and contrasts the work of Iwatsuki-Horimoto et al. Finally, our work supports that of Iwatsuki-Horimoto et al. as neither of our studies detected clinical signs of infection, impaired weight gain, viral RNA, or infectious virus in the lungs of SARS-CoV-2 inoculated guinea pigs, which demonstrates that these findings are replicable by different research groups. We have updated our discussion section to address the work of Iwatsuki-Horimoto et al.
- bioRxiv: https://www.biorxiv.org/content/10.1101/2022.05.20.492834v1.full
- Manuscript: https://pmc.ncbi.nlm.nih.gov/articles/PMC11311394/

Round 2
Reviewer 1 Report
Comments and Suggestions for Authors
I have no more comments. I accepted revision.